# Establishment of a Serum-Free Hepatocyte Cryopreservation Process for the Development of an “Off-the-Shelf” Bioartificial Liver System

**DOI:** 10.3390/bioengineering9120738

**Published:** 2022-11-29

**Authors:** Ji-Hyun Lee, Hey-Jung Park, Young-A Kim, Doo-Hoon Lee, Jeong-Kwon Noh, Jong-Gab Jung, Hee-Hoon Yoon, Suk-Koo Lee, Sanghoon Lee

**Affiliations:** 1Stem Cell and Regenerative Medicine Center, Research Institute for Future Medicine, Samsung Medical Center, Seoul 06351, Republic of Korea; 2Research Institute, HLB Cell Co., Ltd., Hwaseong 18469, Republic of Korea; 3Department of Surgery, Samsung Medical Center, Sungkyunkwan University School of Medicine, Seoul 06351, Republic of Korea

**Keywords:** hepatocytes, hepatocyte spheroids, hepatocyte spheroid beads, cryopreservation, bioartificial liver system

## Abstract

To use hepatocytes immediately when necessary for hepatocyte transplantation and bioartificial liver (BAL) systems, a serum-free cryopreservation protocol ensuring the high survival of hepatocytes and maintenance of their functions should be developed. We established a serum-free protocol for the cryopreservation of primary hepatocytes, hepatocyte spheroids, and hepatocyte spheroid beads in liquid nitrogen. The serum-free cryopreservation solutions showed a significantly higher performance in maintaining enhanced viability and ammonia removal, urea secretion, and the albumin synthesis of hepatocyte spheroids and spheroid beads. The serum-free thawing medium, containing human serum albumin (HSA) and N-acetylcysteine (NAC), was compared with a fetal bovine serum-containing thawing medium for the development of a serum-free thawing medium. Our results show that hepatocyte spheroids and spheroid beads thawed using a serum-free thawing medium containing HSA and NAC exhibited increased hepatocyte viability, ammonia removal, urea secretion, and albumin synthesis compared to those thawed using the serum-containing medium. Finally, we evaluated the liver functions of the cryopreserved BAL system-applied serum-free cryopreservation process compared to the fresh BAL system. The ammonia removal efficiency of the cryopreserved hepatocyte spheroids BAL was lower than or similar to that of the fresh BAL system. Additionally, the urea concentrations in the media of all three BAL systems were not significantly different during BAL system operation. This cryopreserved spheroid-based BAL system using a serum-free process will be a good candidate for the treatment of patients.

## 1. Introduction

Owing to the absolute shortage of liver donors, bioartificial liver (BAL) systems and hepatocyte transplantation serve as useful intermediate solutions by providing metabolic support while patients with liver failure wait for liver transplantation [1,2]. Hepatocyte transplantation can replace liver transplantation in the treatment of certain metabolic liver diseases [3,4]. However, the success of these treatments depends on the quality of the transplanted hepatocytes. The ability to use a large number of hepatocytes immediately, when necessary, in hepatocyte transplantation and BAL systems is one of the ways to maximize the practicality of the approach. For example, in the BAL system, hepatocytes isolated from liver are used. For rapid clinical treatment, it is essential to manufacture a ready-made BAL system that can be provided immediately upon patient admission. 

Cryopreserved hepatocyte spheroids can be thawed and used at any time in an emergency when applied to the BAL system to treat patients with acute liver failure. It can contribute to safety improvement by not only shortening the process time of transporting animals, separating hepatocytes and culturing spheroids, but also pre-testing infectious agents, such as pathogenic bacteria or viruses.

However, when hepatocytes are cryopreserved, cell viability or activity rapidly decreases after thawing. Cryopreserved hepatocytes show a significant decrease in attachment efficiency, cytochrome p450 activity, and protein synthesis [5,6,7,8], and hence are significantly inferior to newly isolated hepatocytes. Therefore, a method for the cryopreservation of hepatocytes that maintains a high level of viability and activity should be developed.

The first complete hepatocyte cryopreservation protocol was published in the 1980s [9,10]; since then, numerous studies to optimize the cryopreservation method have been conducted [11,12,13,14,15,16]. Despite extensive research, the significant loss of hepatocyte viability and function following thawing remains a serious issue, with many of these thawed hepatocytes being unsuitable for clinical use. There are several steps involved in the cryopreservation process that may influence hepatocyte function, including the cryoprotectant used, method and media used for freezing, and thawing process.

Most protocols of cell cryopreservation involve the use of serum in at least one step during freezing, thawing, or downstream processes. While this increases cell viability and recovery following cryopreservation, the adaptation of these protocols for the clinical application of cryopreserved hepatocytes requires the omission of any serum products. When using a cryopreservation solution to which serum is added, it is necessary to purify the cultured cells, resulting in a complicated process, and there is a risk of infection with unknown pathogens such as viruses. Therefore, the risk of infection can be excluded by using a serum-free cryopreservation solution, and safety verification data and procedures can be simplified. That is, it is necessary to develop a serum-free freeze/thaw process.

One of the most critical aspects of the thawing process is the rapid dilution of dimethyl sulfoxide (DMSO), which requires the addition of thawing medium up to 30 times the volume of the cryopreservation solutions and 60–90 L volume is required for BAL application. In addition, when compared to thawing media supplemented with fetal bovine serum (FBS), serum-free thawing media significantly reduces cell viability and liver function activity. Therefore, the serum-free thawing process must be markedly improved to be considered feasible for clinical use.

Cellular isolation may cause cellular trauma, as demonstrated by liver ischemia/reperfusion injuries and the associated impaired mitochondrial function, intracellular ATP depletion, and increased reactive oxygen species (ROS) production, ultimately leading to hepatocyte death [14,15,17]. Based on previous studies, we believe that hepatocyte isolation represents the most significant source of oxidative stress. In a recent hepatocyte cryopreservation study, researchers evaluated a pre-incubation step using antioxidants prior to cellular cryopreservation [18,19,20,21,22]. However, there are no studies regarding the direct use of antioxidants in cryopreservation and thawing processes.

Therefore, we designed this study to evaluate the role of antioxidants in the freeze/thaw process and develop a novel antioxidant-enriched serum-free freeze/thaw medium. In the present study, we evaluate the effects and potential of NAC for clinical application by adding it to a previously developed serum-free cryopreservation solution [23]. Then, the liver function of the cryopreservation BAL system applied with the serum-free freeze/thaw process is evaluated.

## 2. Materials and Methods

If not otherwise stated, reagents were purchased from Sigma Aldrich, St. Louis, MO, USA.

### 2.1. Animals

Male pigs grown in clean barrier facilities (15–22 weeks old, mixed strain of Landrace, Yorkshire, and Duroc weighing 12–18 kg, Optipharm Co. Ltd., Cheongju, Republic of Korea) were used for hepatocyte isolation. This study was reviewed and approved (No. 20180131002) by the Institutional Animal Care and Use Committee (IACUC) of the Samsung Biomedical Research Institute (SBRI) at the Samsung Medical Center (SMC). All experiments described in this section were performed in accordance with the guidelines and regulations of the committee. The SBRI is an Association for the Assessment of the Accreditation of Laboratory Animal Care International (AAALAC International)-accredited facility and abides by the Institute of Laboratory Animal Resources (ILAR) guidelines. The ARRIVE guidelines for reporting animal research were followed [24].

### 2.2. Porcine Hepatocyte Isolation

Hepatocytes were harvested from 15-to-22-week-old male pigs weighing 12–18 kg. Anesthesia was induced with ketamine (20 mg/kg intramuscularly) and xylazine (2 mg/kg intramuscularly). After endotracheal intubation, the ventilator was adjusted to achieve a PCO_2_ between 35 and 40 torr. Anesthesia was maintained with isoflurane (2%) and vecuronium (2 mg/kg/h). Hepatocyte isolation was completed using the three-step collagenase perfusion technique according to the procedures described by Lee et al. [24]. Briefly, the liver was perfused via the portal vein with 8 L of warm oxygenized perfusion buffer at a flow rate of 80 mL/min/kg body weight. Then, 1000 mL of a solution containing calcium (2.5 mM) was introduced through the portal vein to ensure optimal collagenase function. Finally, liver digestion was accomplished by recirculating 1500 mL of a perfusion buffer supplemented with collagenase (0.3 g/L) and calcium chloride (5 mM) at a flow rate of 70 mL/min/kg body weight.

When the liver softened and the capsule began to rupture spontaneously, the collagenase solution was immediately removed, and the digestion process was stopped using 1000 mL of cold (4 °C) Williams E medium. The liver was then cooled in a sterile container on ice, minced, and gradually filtered through a mesh (pore size of 500–150 μm). The cells were then centrifuged, cell pellet was washed with medium, and cells were centrifuged three times at 50× *g* for 4 min at 4 °C. Trypan blue exclusion staining showed that the hepatocytes obtained were >85% viable.

### 2.3. Spheroid Culture and Calcium Alginate Immobilization

Hormonally defined medium (HDM) composed of Williams’ E medium supplemented with 20% albumin (5 mL/L), epidermal growth factor (20 μg/L), insulin (10 mg/L), CuSO_4_·5H_2_O (24.97 μg/L), ZnSO_4_·7H_2_O (14.38 pg/L), H_2_SeO_3_ (3 μg/L), NaHCO_3_ (1.05 g/L), HEPES (1.19 g/L), penicillin (58.8 mg/L), and streptomycin (0.1 g/L) was used to culture the spheroids. The spheroids culture and calcium alginate immobilization procedure were performed using the procedures described by Lee et al. [25]. Briefly, the isolated hepatocytes were inoculated at a cell density of 1.5 × 10^6^ cells/mL in a spinner flask containing 2000 mL of culture medium and stirred with a magnetic stirrer at 20 rpm. The gas phase of the spinner flask was purged with 95% O_2_ and 5% CO_2_ to support the high-oxygen demand of the hepatocytes. These cells were then cultured for 10 h to form spheroids with a mean diameter of 70 μm. Cell viability was assessed by 3-(4, 5-dimethyl thiazol-2-yl)-2, 5-diphenyltetrazolium bromide (MTT) conversion before encapsulation. Viable hepatocyte spheroids were then mixed with a 1.5% alginate solution, which had been previously heat-treated to ensure dissolution and sterilization. This mixture was then placed in a high content/speed immobilization apparatus and dropped into a 100 mM calcium solution.

### 2.4. Procedures of Cryopreservation and Thawing

All steps during the freezing procedure were performed on ice. Primary hepatocytes, primary hepatocyte spheroids, or immobilized hepatocyte spheroid beads were centrifuged at 50× *g* for 5 min at 4 °C, and the supernatant was discarded.

Primary hepatocytes were resuspended in ice-cold cryopreservation solutions, CryoStor CS10 (CS), fetal bovine serum + DMSO (FBS) and modified histidine-tryptophan-ketoglutarate + DMSO (mHTK: consisting of human serum albumin [4%], insulin [3000 U/L], dexamethasone [50 mg/L], and amino acids [0.04%]). The cell solution (1 × 10^7^ cells in 1 mL) was then transferred to cryovials. The vials were then immediately placed into an isopropanol progressive freezing container at −80 °C overnight and immersed in liquid nitrogen after 24 h. One month later, the cryopreserved hepatocytes were rapidly thawed at 37 °C in a water bath. The thawed cell suspension was transferred to a tube containing 30 mL of warmed (37 °C) thawing medium (washing medium supplemented with 10% FBS) and mixed. The resulting cell suspension was centrifuged at 50× *g* for 5 min at 4 °C. The supernatant was discarded, and the cell pellet was resuspended in HDM. Hepatocytes were plated on collagen-coated plastic dishes (5 × 10^5^ cells/mL) and cultured for functional assays after 24 h.

The primary hepatocyte spheroids or spheroid beads were resuspended in ice-cold cryopreservation solutions, FBS, mHTK and mHTK+ 15 mM N-acetylcysteine (mHTK + NAC). All cryopreservation solutions contained as cryoprotectant 15% dimethyl sulfoxide (DMSO). The cell solution (2 × 10^7^ cells in 1 mL) was then transferred to cryovials. Approximately 0.5 mL of beads (3 × 10^7^ cells/mL) were exposed to 1 mL of cryopreservation solution containing 22.5% (*v/v*) DMSO in a freezing solution to achieve a final DMSO concentration of 15%. The vials were then immediately placed into an isopropanol progressive freezing container at −80 °C overnight and immersed in liquid nitrogen after 24 h. One year later, the cryopreserved hepatocyte spheroids and spheroid beads were rapidly thawed at 37 °C in a water bath. The thawed cell suspension and beads were transferred to a tube containing 30 mL of warmed (37 °C) thawing medium (washing medium supplemented with 10% FBS) and mixed. The resulting cell suspension was centrifuged at 50× *g* for 5 min at 4 °C. The supernatant was discarded, and the cell pellet was resuspended in HDM. Hepatocyte spheroids were plated on collagen-coated plastic dishes (5 × 10^5^ cells/mL) and cultured for functional assays after 24 h. For further analyses, the hepatocyte spheroid beads were resuspended in HDM at a cell density of 5 × 10^5^ cells/mL in plastic dishes.

To identify the optimal serum-free thawing medium, the effect of human serum albumin (HAS) and NAC concentrations on the thawing outcomes of cryopreserved cells was evaluated in a pilot study. Hepatocyte spheroids were frozen using an FBS solution containing 15% DMSO. Thawed hepatocyte spheroids were washed in a thawing medium supplemented with 1%, 2%, and 4% HSA (A1, A2, and A4, respectively) or 20, 40, and 50 mM NAC (N2, N4, and N5, respectively) and then evaluated for viability, as described below. The concentrations of HSA and NAC with the best overall outcomes were then used in our downstream experiments.

The thawed hepatocyte spheroids suspensions or beads were transferred to a tube containing 30 mL of warmed (37 °C) thawing medium (10% FBS: washing medium supplemented with 10% FBS or A1N4: washing medium supplemented with 1% HSA and 40 mM NAC) and mixed. The resulting cell suspension was then centrifuged at 50× *g* for 5 min at 4 °C, the supernatant was discarded, and the cell pellet was resuspended in HDM. Hepatocyte spheroids were then plated on collagen-coated plastic dishes (5 × 10^5^ cells/mL) and cultured for 24 h prior to their use in the functional assays. In all further analyses, hepatocyte spheroid beads were resuspended in HDM at a cell density of 5 × 10^5^ cells/mL and cultured in plastic cell culture dishes.

### 2.5. Bioreactor and BAL System

The BAL system was made by scale-down to 1/100 the size of the previously developed BAL system described by Lee et al. [25]. The BAL system consists of a medium reservoir, a pump, an oxygenator, and a cylindrical-type calcium-alginate packed-bed bioreactor containing the immobilized hepatocytes spheroids. The medium enters the BAL system (0.22 mL/min) and recirculates at 5–6 mL/min through the reservoir, the oxygenator/heater, and the bioreactor. After recirculation through the BAL system the medium is removed. A schematic diagram of the BAL system is shown in Figure 1. The total cell number in the BAL system was 1.5 × 10^8^ seeded in bead solution of 5 mL, as shown in the schematic diagram of the BAL operating system. The dissolved oxygen of the medium that passed through the bioreactor was measured and maintained by supplying oxygen so as not to fall below 3 ppm. Oxygen gas comprising 95% oxygen and 5% carbon dioxide was supplied to the reservoir.

### 2.6. Cell Viability Assays

The viability of thawed hepatocytes was determined using the trypan blue dye exclusion methods. The viability and attachment of thawed hepatocyte spheroids were determined using MTT conversion assay. To demonstrate the viability of cells in different cryopreservation/thawing solutions, calcium-alginate beads containing hepatocyte spheroids in the experimental and control groups were assessed using the MTT assay. The Live/Dead viability/cytotoxicity kit from Molecular Probes, Inc. (Eugene, OR, USA) was used to assess live/dead staining in each sample according to the manufacturer’s instructions. Live cells were stained with calcein-AM, which is metabolically converted by intracellular esterases into calcein and a green fluorescent product, while the dead cells were stained with ethidium bromide. Cells were incubated in a medium containing these reagents for 10–15 min at 37 °C, following which they were immediately imaged using an Olympus IX71 instrument (Olympus Co., Tokyo, Japan).

### 2.7. Measurement of Liver-Specific Functions

Ammonia removal and urea production rates were measured by procedures described by Lee et al. [26].

To determine the ammonia removal rate during the culturing of thawed cells, 1 mM NH4Cl was supplemented to the medium, and samples were collected from the changed medium after 1 day of culture. In order to measure the ammonia concentration during BAL operation, the medium was supplemented with 0.3 mM of NH4Cl, and then was circulated. Every 2 h, a sample was collected from the removed medium after circulating. Briefly, 640 µL of sodium tungstate (100 g/L) and 160 µL of the sample were added to a test tube and mixed with 1.0 mL of color reagent I containing phenol (10 g/L), sodium nitroprusside (50 mg/L), and 1.0 mL of color reagent II containing NaOH (5 g/L), Na_2_HPO_4_·12H_2_O (53.6 g/L), and 1% sodium hypochlorite. After gentle vortexing, the tubes were allowed to stand at 37 °C for 20 min, and the samples were analyzed at an absorbance of 630 nm.

To determine the urea production rate, the test tubes were filled with 3 mL of a color reagent produced by mixing 20 mL of diacetyl monoxime (6 g/L) and thiosemicarbazide (0.3 g/L) and 100 mL of 34% H_3_PO_4_, followed by adding 100 µL of the sample. After gentle mixing, the tubes were incubated at 100 °C for 10 min, and the samples were analyzed at an absorbance of 540 nm.

Albumin concentration was determined using a pig albumin ELISA quantitation set (E100–110; Bethyl Laboratories, Montgomery, TX, USA). The results were analyzed using SoftMax Pro 4.8 software (Molecular Devices LLC, San Jose, CA, USA) and fitted with a 4-parameter curve with R^2^ > 0.99.

### 2.8. Statistical Analysis

Data were analyzed using GraphPad Prism^®^ 5 software (GraphPad, San Diego, CA, USA). All data are presented as the mean ± standard deviation from at least 4 independent experiments. Student’s *t*-tests were applied to compare 2 independent groups. Comparisons between 3 or more groups with 1 independent variable were performed using a one-way analysis of variance (ANOVA) and multiple comparisons were adjusted by using Tukey’s post hoc test. A *p*-value of <0.05 was considered statistically significant. A two-way repeated measurement ANOVA and Tukey’s post hoc multiple comparisons tests were used to compare the groups exposed to different conditions.

## 3. Results

### 3.1. Viability and Liver-Specific Functions of Cryopreserved Porcine Hepatocytes

The viability of hepatocytes cryopreserved for 1 month was lower than that of fresh single hepatocytes (FH) (Figure 2A). While the FH showed 100% attachment before freezing, only 30–40% of the cells were attached after the frozen cells were thawed (Figure 2B). The attachment efficiencies of the cryopreserved hepatocytes were 33.9 ± 1.4% (CS), 38.6 ± 0.5% (mHTK), and 41.3 ± 0.6% (FBS), whereas the ammonia removal rates were 21.7 ± 0.3 (CS), 28.5 ± 0.2 (mHTK), and 20.8 ± 0.3 µg/10^6^ cells/day (FBS) (Figure 2C). The urea secretion rates of the cryopreserved hepatocytes were 63.3 ± 3.8 (CS), 114.2 ± 3.2 (mHTK), and 84.6 ± 1.0 µg/10^6^ cells/day (FBS), and their albumin synthesis rates were 491 ± 31 (CS), 1105 ± 88 (mHTK), and 517 ± 18 ng/10^6^ cells/day (FBS) (Figure 2D,E).

The attachment efficiency, ammonia removal, urea secretion and albumin synthesis rate of mHTK were significantly higher than those of CS. The attachment efficiency and urea secretion rate of FBS were significantly higher than those of CS, and the albumin synthesis rate was not significantly different.

The microscopic appearance of the cultured cryopreserved hepatocytes (CS, mHTK, FBS) and non-cryopreserved fresh hepatocytes (FH) was dissimilar (Figure 2F). The FH displayed a polygonal shape, vesicular inclusions, and light-scattering canalicular-like structures at the cell–cell basolateral interfaces of adjacent cells, indicative of the epithelial characteristics of the cells. However, some of the freeze/thawed hepatocytes appeared to be “damaged” and displayed irregular cell shapes (Figure 2F).

### 3.2. Viability and Liver-Specific Functions of Cryopreserved Porcine Hepatocyte Spheroids

The cell viability of the groups cryopreserved for 1 year was similar to that of fresh primary hepatocyte spheroids (FS) (Figure 3A). The attachment efficiencies of the cryopreserved hepatocyte spheroids were lower than or similar to those of the unfrozen FS (Figure 3B) at 86.3 ± 5.7% (FS), 73.8 ± 2.3% (FBS), 80.1 ± 6.3% (mHTK), and 73.9 ± 4.4% (mHTK + NAC). The rates of ammonia removal (Figure 3C) were 25.4 ± 1.1 (FBS), 30.3 ± 0.7 (mHTK), and 27.6 ± 0.7 µg/10^6^ cells/day (mHTK + NAC). The rates of urea secretion (Figure 3D) from the cryopreserved hepatocyte spheroids were 74.8 ± 2.2 (FBS), 86.3 ± 5.5 (mHTK), and 98.3 ± 2.9 µg/10^6^ cells/day (mHTK + NAC), and their rates of albumin synthesis (Figure 3E) were 1185 ± 33 (FBS), 1818 ± 102 (mHTK), and 1783 ± 79 ng/10^6^ cells/day (mHTK + NAC).

The FBS group demonstrated significantly lower ammonia removal than the mHTK and mHTK + NAC groups. Urea secretion in the FBS group was significantly lower than that in the mHTK and mHTK + NAC groups. The albumin synthesis rates in the serum-free cryopreservation groups (mHTK and mHTK + NAC) were significantly higher than those in the FBS cryopreservation group.

The microscopic appearance of the cultured cryopreserved hepatocyte spheroids (FBS, mHTK and mHTK + NAC ) was similar to that of the unfrozen fresh spheroids (FS) (Figure 3F). The cryopreserved hepatocyte spheroids attached to the culture dish within one day after thawing, spreading and culturing as a monolayer. The cells displayed a polygonal shape, vesicular inclusions, and light-scattering canalicular-like structures at the cell–cell basolateral interfaces of adjacent cells, indicative of the epithelial characteristics of the cells.

### 3.3. Viability and Liver-Specific Functions of Cryopreserved Porcine Hepatocyte Spheroid Beads

The cell viability of the groups cryopreserved for 1 year was lower than that of fresh primary hepatocyte spheroid beads (FB) (Figure 4A). The viability of the hepatocyte spheroid beads was determined using the MTT conversion assay after thawing (Figure 4A) as well as calcein-AM and ethidium bromide (EthD-1) staining (Figure 4E). Hepatocyte spheroid beads thawed in 10% FBS thawing medium demonstrated decreased viability, and the results of the MTT assay (Figure 4A) were consistent with those obtained from the staining experiments (Figure 4E).

The ammonia removal rates (Figure 4B) of the cryopreserved hepatocyte spheroid beads were 23.7 ± 3.1 (FBS), 30.7 ± 0.8 (mHTK), and 32.4 ± 0.7 µg/10^6^ cells/day (mHTK + NAC), and their urea secretion rates (Figure 4C) were 60.4 ± 18.8 (FBS), 80.0 ± 3.7 (mHTK), and 91.7 ± 4.9 µg/10^6^ cells/day (mHTK + NAC). The albumin synthesis rates (Figure 4D) were 2490 ± 94 (FBS), 4331 ± 250 (mHTK), and 4715 ± 437 ng/10^6^ cells/day (mHTK + NAC).

The viability and ammonia removal rates of the serum-free cryopreservation groups (mHTK and mHTK + NAC) were significantly higher than those of the FBS cryopreservation group. The FBS group demonstrated significantly lower urea secretion than the mHTK and mHTK + NAC groups. The albumin synthesis rates of the serum-free cryopreservation groups (mHTK and mHTK + NAC) were significantly higher than those of the FBS cryopreservation group.

### 3.4. Optimal Concentration of Human Serum Albumin (HSA) and N-Acetylcysteine (NAC) for Serum-Free Thawing Medium

To identify the optimal serum-free thawing medium, the effect of HSA and NAC concentrations on the thawing outcomes of cryopreserved cells was evaluated in a pilot study. Thawed hepatocyte spheroids were washed in a thawing medium supplemented with 1%, 2%, and 4% HSA (A1, A2, and A4, respectively) or 20, 40, and 50 mM NAC (N2, N4, and N5, respectively).

Following the pilot study, we showed that the post-thaw viability of the hepatocyte spheroids thawed using 4% HSA-containing medium (A4N2) or 40 mM of NAC (A2N4 and A1N4) was similar to that of the spheroids thawed using 10% FBS containing thawing medium (1FBS) (Figure 5A).

The attachment efficiencies of these hepatocyte spheroids were 57.6 ± 4.1% (1FBS), 57.1 ± 3.2% (A4N2), 65.3 ± 3.2% (A2N4), 70.3 ± 4.5% (A1N4), and 62.9 ± 2.4% (A1N5). The attachment efficiency was significantly higher in the 40 mM (A2N4 and A1N4) and 50 mM NAC thawing medium groups (A1N5) than in the F group (Figure 5B).

The ammonia removal (29.1 ± 0.3 and 28.9 ± 0.1 µg/10^6^ cells/day, Figure 5C), urea secretion (90.0 ± 1.1 and 89.4 ± 3.3 µg/10^6^ cells/day, Figure 5D), and albumin synthesis (211 ± 13 and 211 ± 7.9 ng/10^6^ cells/day, Figure 5E) assays all produced the highest values in the 40 mM NAC group (A2N4 and A1N4), and these parameters were significantly higher than those in the 1FBS group. Urea secretion (Figure 5D) and albumin synthesis (Figure 5E) were also significantly higher in the 40 mM NAC group (A2N4 and A1N4) than in the 4% HSA group (A4N2).

As there were no differences in hepatocyte spheroid liver-specific functions between the 1%, 2%, and 4% HSA groups, we used 1% HSA in the downstream assays to attempt to improve the economic functions of this approach. In addition, 40 mM of NAC contributed to the most significant improvements in the viability and functional assays; thus, this concentration was selected for the subsequent thawing experiments.

### 3.5. Viability and Liver-Specific Functions of Porcine Hepatocyte Spheroids Thawed Using Serum-Free Medium

After 1 year, cryopreserved hepatocyte spheroids using FBS and mHTK were thawed with 10% FBS thawing medium and A1N4 thawing medium, respectively.

The cell viability of the four thawed groups was similar (Figure 6A). Furthermore, the attachment efficiency of the hepatocyte spheroids thawed using the A1N4 thawing medium increased compared to that of the other groups (Figure 6B). The attachment efficiencies of the hepatocyte spheroids thawed using 10% FBS-containing thawing medium were 73.9 ± 2.3% (FBS) and 80.1 ± 6.4% (mHTK), while the attachment efficiencies of the hepatocyte spheroids thawed using A1N4 thawing medium were 91.1 ± 3.5% (FBS) and 89.7 ± 4.8% (mHTK). The attachment efficiencies of the serum-free thawing medium groups (A1N4) were significantly higher than those of the 10% FBS group (Figure 6B).

The ammonia removal (30.4 ± 0.8 vs. 31.6 ± 0.5 µg/10^6^ cells/day), urea secretion (86.3 ± 5.5 vs. 94.3 ± 3.4 µg/10^6^ cells/day), and albumin synthesis (1818 ± 102 vs. 2516 ± 158 ng/10^6^ cells/day) assays yielded the highest values for the mHTK group (10% FBS vs. A1N4), and these parameters were significantly higher than those in the FBS cryopreservation group (Figure 6C–E). Importantly, the attachment efficiency, ammonia removal, urea secretion, and albumin synthesis in the FBS/A1N4 (freeze/thaw) group were significantly higher than those in the FBS/10% FBS (freeze/thaw) group.

### 3.6. Viability and Liver-Specific Functions of Porcine Hepatocyte Spheroid Beads Thawed Using Serum-Free Medium

After 1 year, cryopreserved hepatocyte spheroid beads using FBS and mHTK were thawed with 10% FBS thawing medium and A1N4 thawing medium, respectively.

The viability of the hepatocyte spheroid beads was determined using the MTT conversion assay after thawing (Figure 7A) as well as calcein-AM and ethidium bromide (EthD-1) staining (Figure 7E). The cell viability of the serum-free thawing groups (A1N4) showed significantly higher cell viability than the thawing group with 10% FBS-containing medium, and the results of the MTT assay (Figure 7A) are consistent with those obtained in the staining experiments (Figure 7E). The viability of the hepatocyte spheroid beads thawed using 10% FBS-containing medium was 36.3 ± 2.3% (FBS) and 58.0 ± 1.5% (mHTK), while the viability of those thawed using A1N4-containing medium was 63.9 ± 9.6% (FBS) and 76.7 ± 5.9% (mHTK).

The ammonia removal rate increased in the serum-free thawing groups (A1N4) compared to the thawing group with 10% FBS containing medium (Figure 7B). The urea secretion of the hepatocyte spheroid beads thawed in 10% FBS-containing medium was 60.4 ± 18.8 µg/10^6^ cells/day (FBS) and 80.0 ± 3.7 µg/10^6^ cells/day (mHTK), while the urea secretion of the hepatocyte spheroid beads thawed using A1N4-containing medium was 68.7 ± 14.4 µg/10^6^ cells/day (FBS) and 98.1 ± 9.3 µg/10^6^ cells/day (mHTK) (Figure 7C). Furthermore, the albumin synthesis of the hepatocyte spheroid beads thawed using 10% FBS-containing medium was 2490 ± 94 ng/10^6^ cells/day (FBS) and 4331 ± 250 ng/10^6^ cells/day (mHTK), while the albumin synthesis of the hepatocyte spheroid beads thawed in A1N4-containing medium was 2635 ± 912 ng/10^6^ cells/day (FBS) and 5700 ± 579 ng/10^6^ cells/day (mHTK) (Figure 7D). The albumin synthesis levels of the serum-free cryopreservation and thawing groups (mHTK/A1N4) were significantly increased compared other groups.

### 3.7. Liver-Specific Functions of Cryopreserved Porcine Hepatocyte Spheroids in a BAL System

In this study, a small prototype of 1/100 of the BAL system was used. The packed-bed bioreactor contained Ca-alginate-immobilized hepatocyte spheroid beads. The BAL (CS-BAL), in which porcine hepatocyte spheroids were cryopreserved and thawed after 2 months to immobilize beads, and the BAL (CB-BAL), in which porcine hepatocyte spheroid beads were cryopreserved and thawed after 2 months, were compared to the fresh BAL.

The ammonia removal efficiency of the CS-BAL was lower than or similar to the fresh BAL system (Figure 8A). The ammonia removal efficiency of CB-BAL dropped sharply after 6 h of BAL operation and was very low compared to fresh BAL and CS-BAL systems.

However, the urea concentrations in the media of all three BAL systems were not significantly different during BAL system operation (Figure 8B).

During 24 h BAL operation, the mean ammonia removal efficiencies (Figure 8A) were 98.3 ± 4.1 (fresh BAL), 92.1 ± 5.7 (CS-BAL), and 37.2 ± 20.2 % (CB-BAL), and their mean urea concentrations (Figure 8B) were 60.0 ± 15.8 (fresh BAL), 49.2 ± 16.0 (CS-BAL), and 40.0 ± 13.1 % (CB-BAL). The mean albumin concentrations (Figure 8C) were 1533 ± 541 (fresh BAL), 368 ± 120 (CS-BAL), and 294 ± 118 % (CB-BAL).

## 4. Discussion

The aim of this study was to develop a freezing/thawing protocol that allows the delivery of hepatocyte spheroids or immobilized hepatocyte spheroids with high functional recovery when required for the BAL treatment of acute liver failure. While a large number of studies have described the cryopreservation of single cells, including hepatocytes, limited information is currently available regarding the cryopreservation of spheroid types or immobilized cells.

Hepatocytes are easily damaged by the freeze–thaw process [27]. Several studies have specifically discussed the cryopreservation of animal hepatocytes for application in BAL. Nyberg et al. [28] have reported that, when comparing albumin production per cell between fresh and cryopreserved cells, the production of cryopreserved cells was 2% of that of the fresh cells. Our albumin synthesis assay results are consistent with those of other studies [29,30,31]. In this study, albumin synthesis by the cells cultured post-thaw was particularly low. A comparison of the per cell albumin production between the fresh and cryopreserved cells revealed that albumin production by hepatocytes cryopreserved using FBS was reduced to 6.5% of that of the non-frozen hepatocytes and 18% of that of the cells cryopreserved with mHTK.

While FBS is commonly used in cryopreservation protocols, its xenogenous origin is disadvantageous in clinical applications, rendering these methods unsuitable for use in humans. Since it is difficult to ensure the complete removal of FBS from the cryopreserved samples, its use is relatively rare, and most methods rely on the use of allogeneic serums [32]. Human serum may be an allogeneic alternative to FBS due to the presence of HSA, which exhibits a serum-like protective effect [33]. Therefore, we used HSA throughout the thawing process to investigate its effects further. However, the use of HSA in high-throughput systems is prohibitively expensive, suggesting that this may not be an economically viable approach in clinical settings. The production of “off-the-shelf” cell therapy products is likely to require a batch-type approach to both cryopreservation and the thawing of large quantities of cells. There are only 1–2 × 10^11^ hepatocytes in a normal adult human liver, and functionally active hepatocytes are required for the repair of damaged liver tissues in patients with acute liver failure. Surgical resection data suggest that the functional reconstitution of the liver requires the application of between 10% and 30% of the liver mass in healthy cells, corresponding to 150–450 g of cells [34]. Solutions with 2–3 × 10^10^ frozen and preserved hepatocytes would have a volume of 1000–1500 mL, and 30–45 L of thawing medium would be required (1:30) to properly dilute the cryopreservation reagent (DMSO). Additionally, 2–3 × 10^10^ hepatocyte spheroid beads would constitute a volume of 2–3 L and require 60–90 L of thawing medium for resuscitation. Therefore, in addition to the reduction in HSA during thawing, we focused on developing a thawing medium that increased hepatocyte viability and liver-specific function following the freeze/thaw process.

Oxidative stress is an injury-causing mechanism that occurs during cryopreservation and results in decreased cell viability and function. Hepatocytes undergo oxidative stress during cryopreservation, but its effects can be reduced by using antioxidants [35]. However, there are no studies regarding the direct use of antioxidants in hepatocyte cryopreservation and thawing processes. NAC is an antioxidant that acts as a hepatic glutathione store replenisher exerting direct antioxidant effects [36]. We identified NAC as a suitable supplement for these media and showed that its addition reduced the amount of HSA required to maintain cell viability. NAC was added to the cryopreservation solution to evaluate its antioxidant effect. The serum-free-modified HTK solutions (mHTK and mHTK + NAC) developed in this study showed significantly higher performance in terms of maintaining the viability and liver-specific functions of the hepatocytes, compared to FBS. The addition of NAC (mHTK + NAC) to the cryopreservation solution significantly increased the ability of the hepatocytes, hepatocyte spheroids, and hepatocyte spheroid beads to secrete urea. NAC exerts its hepatoprotective effects through several different mechanisms [1]. In addition, in vivo data suggest that NAC may protect against liver ischemia/reperfusion injury [37,38], and the use of NAC improves the viability of human hepatocytes isolated from steatotic livers [39]. The hepatoprotective functions of NAC activity may be explained by its capacity to prevent ROS-associated cell damage. The pre-incubation of hepatocytes in a medium supplemented with sugars, insulin, reduced glutathione, and NAC prior to cryopreservation improves cell recovery after thawing [40]. Recent studies have investigated the effects of NAC in the pre-incubation stages of hepatocyte cryopreservation [40]; however, only a few studies have examined its role in the hypothermic preservation of hepatocytes and their cold storage [41].

After hepatocyte spheroid cryopreservation using FBS freezing medium or serum-free mHTK medium, the addition of NAC during the thawing process significantly increased cell viability and liver function compared to the hepatocyte spheroids thawed using 10% FBS-containing medium. However, unlike the beneficial effects of NAC on spheroid cryopreservation, the addition of NAC during the thawing process for hepatocyte spheroid beads did not significantly improve hepatic functional activity. The cryopreservation of hepatocyte spheroid beads has the advantage of providing an off-the-shelf product, as the cells are cryopreserved in the same format for use in clinical settings. The addition of NAC during the thawing process for hepatocyte spheroid beads stored in serum-free mHTK medium significantly improved cell viability, ammonia removal, urea secretion, and albumin synthesis compared to hepatocyte spheroid beads thawed using 10% FBS-containing thawing medium. The thawing of the serum-free solution may take longer in the beads because the medium needs to penetrate both the alginate layer and spheroid. This may be the reason for the reduced improvements in hepatic functional activity in these samples compared to those in the regular spheroids. Our results demonstrate that most beads retain their pre-cryopreservation shapes. However, we also observed that a minority of hepatocyte spheroid beads broke following the freeze–thaw process. Therefore, further research regarding bead tolerance to cryopreservation is necessary before clinical application.

In conclusion, the serum-free cryopreservation process developed in this study improved the viability and functions of thawed cells and provided a method for obtaining cryopreserved hepatocytes of a superior quality for clinical applications. Additionally, the use of a serum-free thawing medium supplemented with NAC can provide an economical method for the mass thawing process required for the clinical application of BAL. Thus, this cryopreserved spheroids-based BAL system using a serum-free process will be a good candidate for the treatment of patients.

## Figures and Tables

**Figure 1 bioengineering-09-00738-f001:**
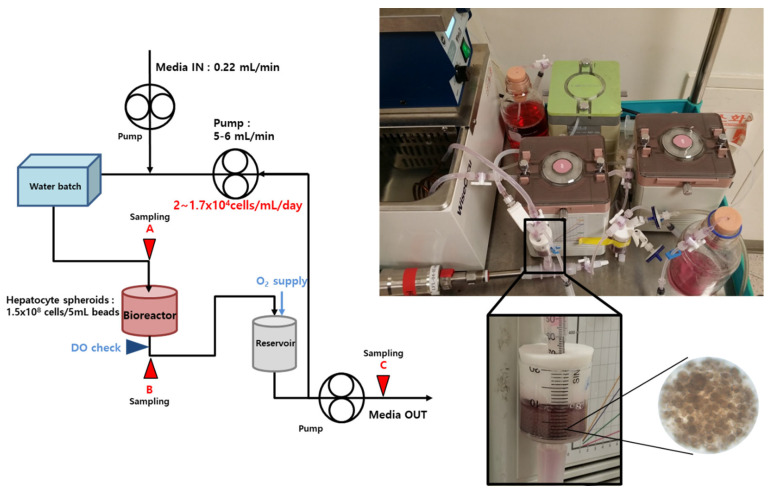
Schematic diagram of the in vitro BAL operating system and photographs of BAL system with immobilized hepatocyte spheroids. New medium flow was supplied to the system and later removed. The medium passes through the BAL system and recirculates at 5–6 mL/min through the reservoir, a heater, and the bioreactor. After recirculation through the BAL system, the medium is removed by the pump.

**Figure 2 bioengineering-09-00738-f002:**
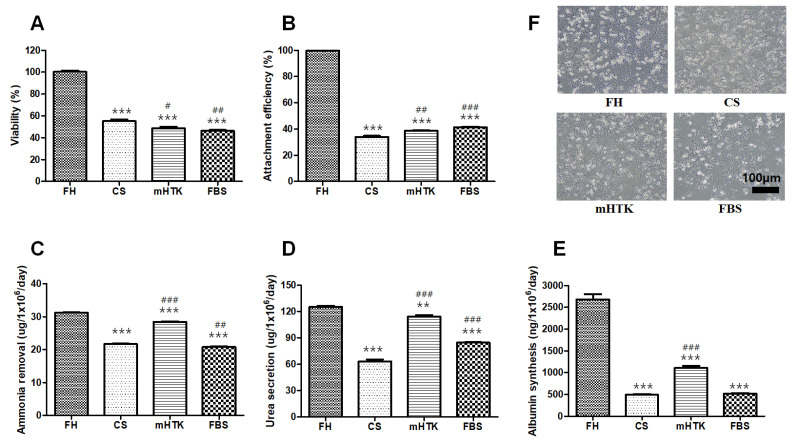
Effect of cryopreservation solutions on porcine hepatocytes viability and liver-specific functions in post-thaw cultures. Porcine hepatocytes were cryopreserved in three different freezing solutions containing 15% DMSO and stored for 1 month in liquid nitrogen. The hepatocytes were thawed by rapid warming in a water bath and washed twice in thawing medium (first, the cells were rapidly suspended in a warm medium with 10% FBS, and then the medium was replaced with a warm HDM). (**A**) Cell viability, (**B**) attachment efficiency (assessed using a 3-(4,5-dimethyl thiazol-2-yl)-2,5-diphenyltetrazolium bromide assay), (**C**) ammonia removal, (**D**) urea secretion, and (**E**) albumin synthesis were evaluated after thawing and plating for 24 h. Representative images of (**F**) porcine hepatocytes showing their morphology after plating and maintenance in culture for 24 h (×100). Fresh primary hepatocytes (upper left, FH) and hepatocytes cryopreserved using CryoStor CS10 (upper right, CS), modified HTK + DMSO (lower left, mHTK), and fetal bovine serum + DMSO (lower right, FBS). Statistical significance compared to fresh primary hepatocytes (FH): ** *p* < 0.01, *** *p* < 0.001, and compared to the control freezing solution (CS): # *p* < 0.05, ## *p* < 0.01, ### *p* < 0.001.

**Figure 3 bioengineering-09-00738-f003:**
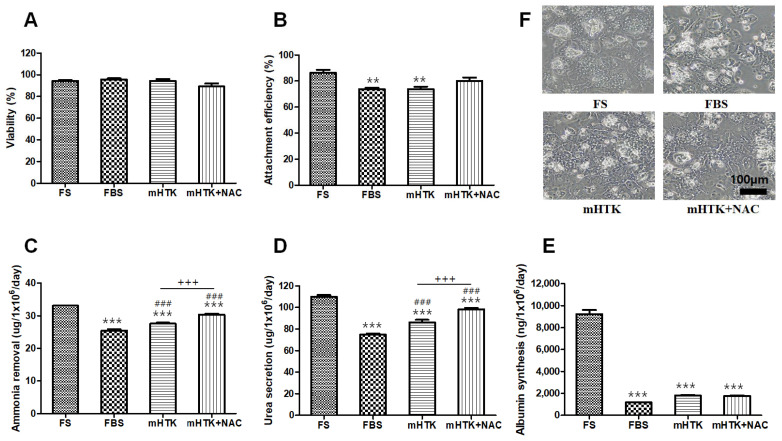
Effects of cryopreservation solutions on porcine hepatocyte spheroids’ viability and liver-specific functions in post-thaw cultures. Porcine hepatocyte spheroids were cryopreserved in three different freezing solutions containing 15% DMSO and stored for 1 year in liquid nitrogen. Porcine hepatocyte spheroids were thawed by rapid warming in a water bath and washed twice in thawing medium (first, hepatocyte spheroids were suspended in a warm medium with FBS rapidly, and then the medium was replaced with a warm HDM). (**A**) Cell viability, (**B**) attachment efficiency (assessed using a 3-(4,5-dimethyl thiazol-2-yl)-2,5-diphenyltetrazolium bromide assay), (**C**) ammonia removal, (**D**) urea secretion, and (**E**) albumin synthesis were evaluated after thawing and plating for 24 h. Representative images of (**F**) porcine hepatocyte spheroids showing their morphology after plating and maintenance in culture for 24 h (×100). Fresh hepatocyte spheroids (upper left, FS) and hepatocytes cryopreserved using fetal bovine serum + DMSO (upper right, FBS), modified HTK + DMSO (lower left, mHTK), and modified HTK + N-acetylcysteine + DMSO (lower right, mHTK + NAC). Statistical significance compared to fresh primary hepatocyte spheroids (FS): ** *p* < 0.01, *** *p* < 0.001, and compared the control freezing solution (FBS): ### *p* < 0.001. Statistical significance: +++ *p* < 0.001.

**Figure 4 bioengineering-09-00738-f004:**
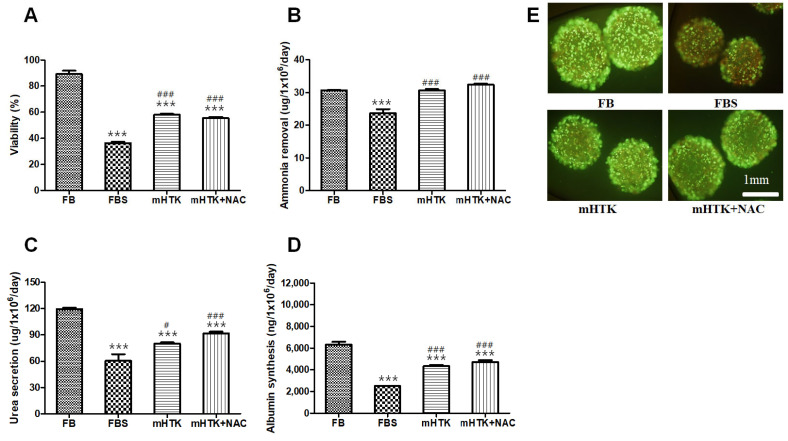
Effect of cryopreservation solutions on porcine hepatocyte spheroid beads’ viability and liver-specific functions in post-thaw cultures. Porcine hepatocyte spheroid beads were cryopreserved in three different freezing solutions containing 15% DMSO and stored for 1 year in liquid nitrogen. Porcine hepatocyte spheroid beads were warmed rapidly in a water bath and washed twice in thawing medium (first, hepatocyte spheroid beads were suspended in a warm medium with 10% FBS rapidly, and then the medium was replaced with a warm HDM). (**A**) Cell viability, (**B**) ammonia removal, (**C**) urea secretion, and (**D**) albumin synthesis were evaluated after thawing and plating for 24 h. Representative images of (**E**) porcine hepatocyte spheroid beads showing their vital staining after thawing and maintenance in culture for 24 h. Live/Dead^TM^ cell viability assay showing live cells stained with Calcein-AM (green) and dead cells stained with EthD-1 (red). Fresh hepatocyte spheroid beads (upper left, FB) and hepatocyte spheroid beads cryopreserved using fetal bovine serum + DMSO (upper right, FBS), modified HTK + DMSO (lower left, mHTK), and modified HTK + N-acetylcysteine + DMSO (lower right, mHTK + NAC). Statistical significance compared to fresh primary hepatocyte spheroid beads (FB): *** *p* < 0.001 and compared to the control freezing solution (FBS): # *p* < 0.05, ### *p* < 0.001.

**Figure 5 bioengineering-09-00738-f005:**
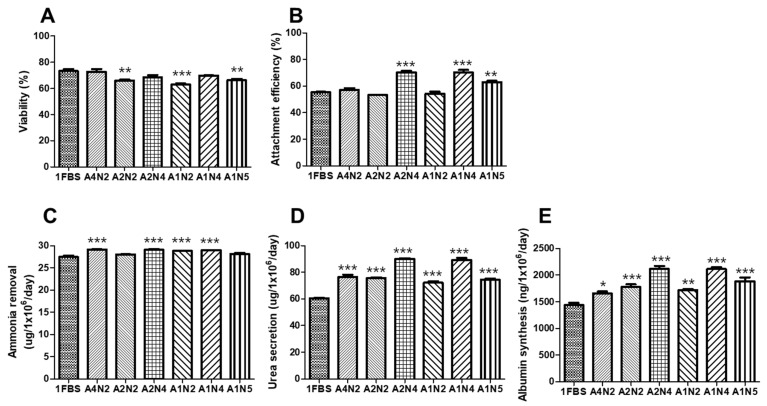
Effect of thawing medium on cryopreserved porcine hepatocyte spheroids’ viability and liver-specific functions in post-thawing culture. Porcine hepatocyte spheroids were cryopreserved in FBS supplemented with 15% DMSO and stored for 1 year in liquid nitrogen. Porcine hepatocyte spheroids were warmed rapidly using a water bath and washed in a thawing medium supplemented with 1%, 2%, and 4% human serum albumin (A1, A2, and A4, respectively) or 20, 40, and 50 mM N-acetylcysteine (N2, N4, and N5, respectively) and then resuspended in warm complete medium and evaluated for (**A**) cell viability; (**B**) attachment efficiency using a (3-(4,5-dimethyl thiazol-2-yl)-2,5-diphenyltetrazolium bromide assay; (**C**) ammonia removal; (**D**) urea secretion; and (**E**) albumin synthesis after 24 h of culture. 1FBS, 10% FBS; A, human serum albumin; N, N-acetylcysteine. Statistical significance compared to the control thawing medium (1FBS): * *p* < 0.05, ** *p* < 0.01, *** *p* < 0.001.

**Figure 6 bioengineering-09-00738-f006:**
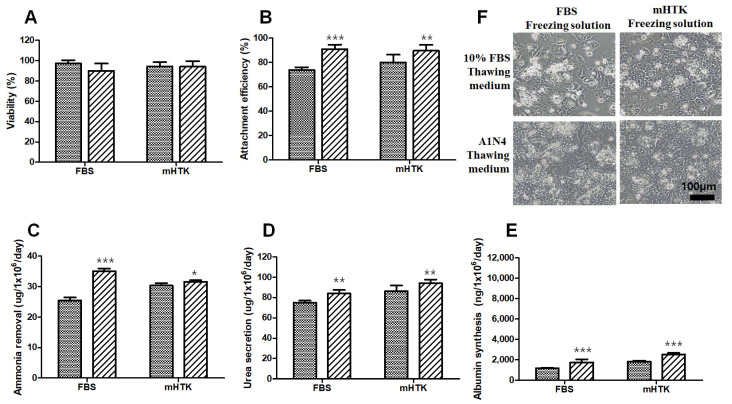
Effect of freezing and thawing solutions on porcine hepatocyte spheroids’ viability and liver-specific functions in post-thawing culture. Porcine hepatocyte spheroids were cryopreserved in two different freezing solutions supplemented with 15% DMSO and stored for 1 year in liquid nitrogen. Porcine hepatocyte spheroids were warmed rapidly using a water bath and washed twice in two different thawing solutions (a warm serum medium (10% FBS, ▩) or serum-free medium (A1N4, ▨)) and then resuspended in a warm complete medium and evaluated for (**A**) cell viability; (**B**) attachment efficiency using a (3-(4,5-dimethyl thiazol-2-yl)-2,5-diphenyltetrazolium bromide assay; (**C**) ammonia removal; (**D**) urea secretion; and (**E**) albumin synthesis after 24 h of culture. Representative images of (**F**) porcine hepatocyte spheroids showing their morphology after plating and maintenance in culture for 24 h. Cryopreserved hepatocytes using FBS then thawed using 10% FBS thawing medium (upper right) and A1N4 (lower left) or cryopreserved hepatocytes using mHTK and then thawed using 10% FBS thawing medium (upper right) and A1N4 (lower right). Statistical significance compared to the control thawing medium (10% FBS): * *p* < 0.05, ** *p* < 0.01, *** *p* < 0.001.

**Figure 7 bioengineering-09-00738-f007:**
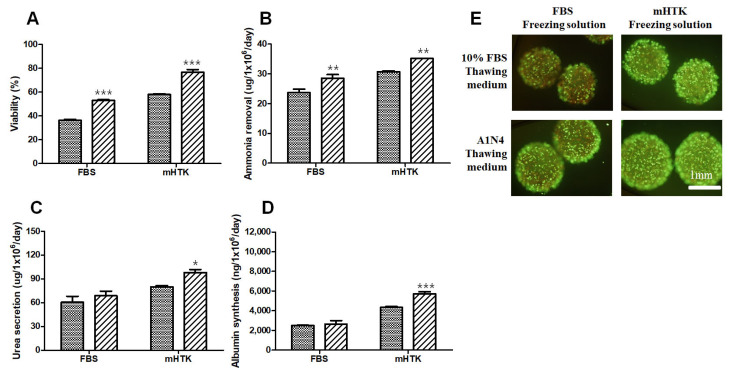
Effects of freezing and thawing solutions on the viability and liver-specific functions of porcine hepatocyte spheroid beads in the post-thawing culture. Porcine hepatocyte spheroid beads were cryopreserved in two different freezing solutions supplemented with 15% DMSO and stored for 1 year in liquid nitrogen. Porcine hepatocyte spheroid beads were warmed rapidly using a water bath and washed twice in a thawing medium (warm serum medium (10% FBS, ▩) or serum-free medium (A1N4, ▨)). They were then resuspended in warm complete medium and evaluated for (**A**) cell viability; (**B**) ammonia removal; (**C**) urea secretion; and (**D**) albumin synthesis after 24 h of culture. Representative images of (**E**) porcine hepatocyte spheroid beads showing their vital staining after thawing and maintenance in culture for 24 h. Live/Dead^TM^ cell viability assay showing live cells stained with Calcein-AM (green) and dead cells stained with EthD-1 (red). Cryopreserved hepatocytes using FBS then thawed using 10% FBS thawing medium (upper left) A1N4 (lower left) or cryopreserved hepatocytes using mHTK then thawed using 10% FBS thawing medium (upper right) and A1N4 (lower right). Statistical significance compared to the control thawing medium (10% FBS): * *p* < 0.05, ** *p* < 0.01, *** *p* < 0.001.

**Figure 8 bioengineering-09-00738-f008:**
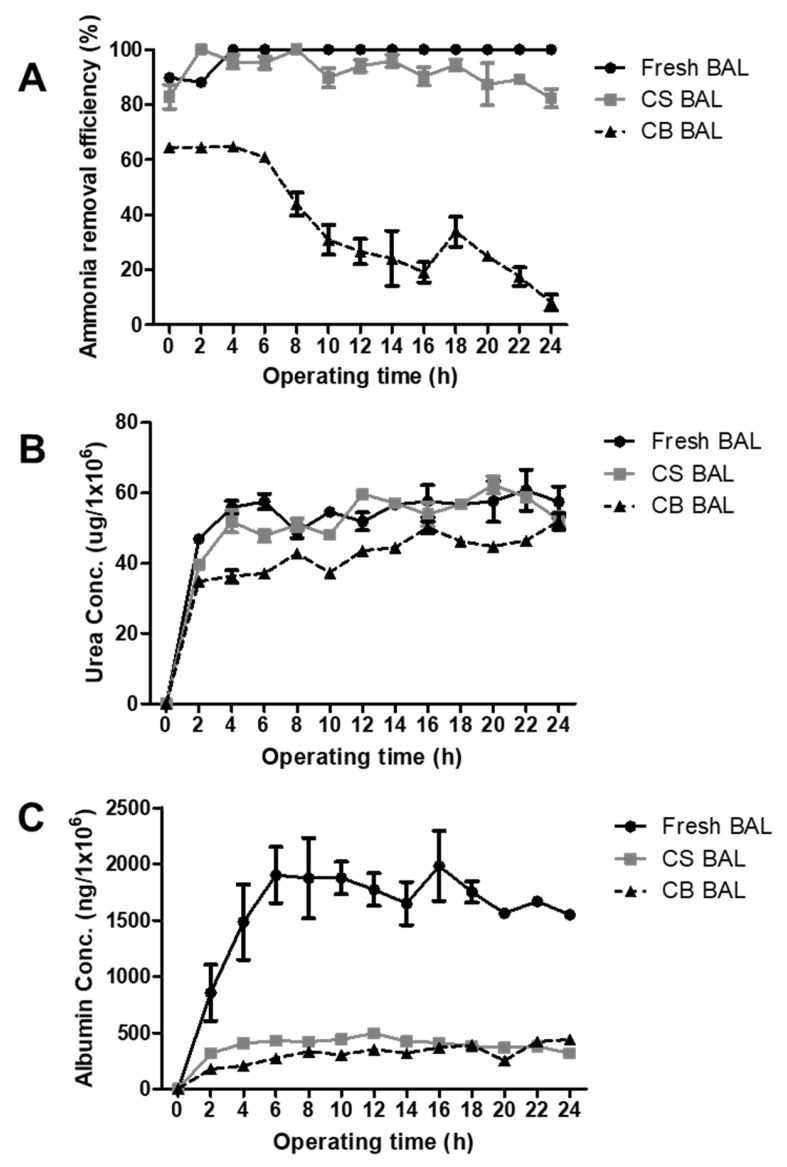
The functions of ammonia clearance, urea synthesis, and albumin synthesis in the bioartificial liver (BAL) system during 24 h operation. (**A**) The ammonia removal efficiency of the CS-BAL was lower than or similar to the fresh BAL system (*p* > 0.05). (**B**) The urea concentrations in the media of the three BAL systems were not significantly different (*p* > 0.05), and (**C**) the albumin concentration in the medium of fresh spheroid bead BAL was higher than that of the cryopreserved BAL (CS BAL, CB BAL) (*p* < 0.05).

## Data Availability

All data from this study are available from the authors upon request.

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
