# Peer review of "Establishment of a Serum-Free Hepatocyte Cryopreservation Process for the Development of an “Off-the-Shelf” Bioartificial Liver System"

_bioengineering, 2022, doi:10.3390/bioengineering9120738_

Round 1

Reviewer 1 Report

The study by Lee J-H. et al. established a serum-free protocol for cryopreservation of primary hepatocytes, hepatocyte spheroids, and hepatocyte spheroid beads in liquid nitrogen. The study clearly shows that hepatocyte spheroids and spheroid beads thawed using serum-free thawing medium containing human serum albumin and N-acetylcysteine exhibited increased hepatocyte viability, ammonia removal, urea secretion, and albumin synthesis. Finally, they suggested that the use of a serum-free thawing medium supplemented with N-acetylcysteine could provide an economical method for the mass thawing process required for the clinical application of BAL. The introduction provides sufficient background and the methods are adequately described. However, I wonder why the authors ignored the very important parameters for NAC-mediated antioxidant activity such as ROS Production, GSH and Total Antioxidant capacity. Apart from this subject, the study planning and writing style is very clear and I think it will guide both methodological and future studies for the reader.

Minor comments:

1. Line 280 should follow Line 279 and belong to the figure legend 2 “and fetal bovine serum+DMSO (FBS). Statistical significance compared to fresh…”

2. Line 283: “The attachment efficiency, ammonia removal, urea secretion and albumin synthesis rate were significantly higher than that of CS. attachment efficiency and urea secretion rate of FBS were significantly higher than those of CS, and albumin synthesis rate was not  significantly different.” It is not clear which group we are talking about.

3. Line 316-318 should follow Line 315.

4. Line 349-354 should follow Line 348.

5. Line 388-389 should follow Line 387.

6. Line 421-434 should follow Line 420.

7. Line 468-469 should follow Line 467.

Reviewer 2 Report

Comments to the Author

General comments to the Authors

Despite the positive reviews of the original versions of the manuscript, there are glaring weaknesses that significantly diminish enthusiasm for its potential clinical utility in hepatocyte transplantation and bioartificial liver. First, the study lacks requisite statistical power and replication to reliably validate the accuracy and reproducibility of its results and conclusions. Second, the study is largely confirmatory of a previously published study by J Biosci Bioeng. 2014 Jul;118(1):101-6.; Sci Rep. 2017 Jul 20;7(1):6080.; Hepatology. 2003 Nov;38(5):1095-106.; and Cell Transplant. 2012;21(10):2257-66. therefore lacks significant novelty.

Reviewer 3 Report

1. There should be a conclusion section in the manuscript.

2. The number of references in the introduction is too less for this important manuscript.

3. The materials section is too simple.

Round 2

Reviewer 2 Report

Comments to the Author

Second review report for a manuscript entitled “Establishment of serum-free hepatocyte cryopreservation process for development of "off-the-shelf" bio-artificial liver system” submitted by the first author Ji-Hyun Lee and corresponding authors Sanghoon Lee. The authors ordered many positive data fitting their advocation. However, this study is too immature, and many data are scientifically questionable.

1.          Several growth factors have been implicated in early liver development and hepatocyte differentiation. The author should further analyzed the expression of hepatocyte markers such as the fetal hepatocytes expressed alpha-fetoprotein (AFP), cytokeratin (CK) 18 & 19, alpha1-AT, albumin, HNF4, HNF3alpha and other hepatic proteins in bio-artificial liver (BAL) systems.

2.          The "Discussion" section needs to be re-written. It is important to "discuss" various issues but not to “re-describe" the findings in this section.

3.          Please correct all of the grammatical mistakes present in the manuscript.

Reviewer 3 Report

Can be accepted.

Author Response

We would like to take this opportunity to express our thanks to the reviewers for the positive feedback and appreciate the time and effort the reviewers have dedicated to providing insightful feedback on ways to strengthen our paper.

Round 3

Reviewer 2 Report

All suggestions has completed and scrutinized. About my reviews; authors explained the all steps.  It can be accept. It has sufficed.